HDAC inhibition delays photoreceptor loss in Pde6b mutant mice of retinitis pigmentosa: insights from scRNA-seq and CUT&Tag

Dong Yujie 1 2
Yan Jie 2 3
Xu Wenrong 2
Paquet-Durand François 3
Hu Zhulin Hzl77@263.net 2
Jiao Kangwei kangwei.jiao@ynu.edu.cn 2
1 Kunming Medical University , Kunming , Yunnan , China
2 Key Laboratory of Yunnan Province, Yunnan Eye Institute, Affiliated Hospital of Yunnan University, Yunnan University , Kunming , Yunnan , China
3 Institute for Ophthalmic Research, Eberhard-Karls-Universität Tübingen , Tübingen , Germany
Dong Peixin
Electronic publication date: 2023 Jul 12
Publication date: 2023
Volume: 11
Electronic Location ID: e15659
Received 2023 Apr 26; Accepted 2023 Jun 7
Copyright: ©2023 Dong et al.
Copyright year: 2023
Copyright holder: Dong et al.
License: This is an open access article distributed under the terms of the Creative Commons Attribution License, which permits unrestricted use, distribution, reproduction and adaptation in any medium and for any purpose provided that it is properly attributed. For attribution, the original author(s), title, publication source (PeerJ) and either DOI or URL of the article must be cited.
License URL: https://creativecommons.org/licenses/by/4.0/

Keywords: Retinitis pigmentosa, Hereditary retinal neurodegenerative disorders, Photoreceptor cells, Histone deacetylases, Poly (ADP-ribose) polymerases, Cell death mechanisms

Funding: National Natural Science Foundation of China 82260213 81960180 Key Project of Yunnan Fundamental Research Projects 202301AS070046 Medical Leading Talents Training Program of Yunnan Provincial Health Commission L-2019029 Joint Project of Yunnan Provincial Department of Science and Technology, Kunming Medical University on Applied Basic Research 202001AY070001-165 202301AY070001-184 Scientific Research Fund of Education Department of Yunnan Province 2023J0050 2023J0049 This research was funded by the National Natural Science Foundation of China (No. 82260213, 81960180), the Key Project of Yunnan Fundamental Research Projects (202301AS070046), the Medical Leading Talents Training Program of Yunnan Provincial Health Commission (L-2019029), the Joint Project of Yunnan Provincial Department of Science and Technology, Kunming Medical University on Applied Basic Research (No. 202001AY070001-165, 202301AY070001-184), and the Scientific Research Fund of Education Department of Yunnan Province (No. 2023J0050, 2023J0049). The funders had no role in study design, data collection and analysis, decision to publish, or preparation of the manuscript.

==============================
Purpose

This research aimed to ascertain the neuroprotective effect of histone deacetylase (HDAC) inhibition on retinal photoreceptors in Pde6brd1 mice, a model of retinitis pigmentosa (RP).

Methods

Single-cell RNA-sequencing (scRNA-seq) explored HDAC and poly (ADP-ribose) polymerase (PARP)-related gene expression in both Pde6b-mutant rd1 and wild-type (WT) mice. The CUT&Tag method was employed to examine the functions of HDAC in rd1 mice. Organotypic retinal explant cultures from WT and rd1 mice were exposed to the HDAC inhibitor SAHA (suberoylanilide hydroxamic acid) postnatally, from day 5 to day 11. The terminal deoxynucleotidyl transferase-mediated nick-end labeling (TUNEL) assay was applied to quantify the percentage of photoreceptor loss in the outer nuclear layer (ONL). HDAC activity was confirmed to be inhibited by SAHA through an HDAC activity assay. Moreover, the study evaluated PARP activity, a key driver of the initial response to DNA damage during photoreceptor degeneration, following HDAC inhibition.

Results

The scRNA-seq revealed that diverse roles of HDAC and PARP isoforms in photoreceptor cell death. HDAC-related genes appeared to regulate cell death and primary immunodeficiency. Alterations in HDAC activity were consistent with the TUNEL-positive cells in the ONL at different time points. Notably, SAHA significantly postponed photoreceptor loss and decreased HDAC and PARP activity, thereby implicating both in the same degenerative pathway.

Conclusions

This study highlights that the interaction between HDAC inhibition and PARP can delay photoreceptor cell death, proposing a promising therapeutic approach for RP.

Introduction

Retinitis pigmentosa (RP) encompasses a group of inherited retinal neurodegenerative disorders (IRDs) characterized by a progressive decline in photoreceptor cell survival (O’Neal & Luther, 2022). The progression of RP begins with rod cells, leading to compromised night vision and a gradual reduction of peripheral vision in the early stage. This is followed by cone cell degeneration, eventually resulting in total blindness (Power et al., 2020c). RP impacts approximately 1 in every 4,000 individuals (Bertelsen et al., 2014). At present, over 90 gene mutations associated with RP have been identified, affecting the survival of retinal photoreceptor or pigment epithelial cells (https://web.sph.uth.edu/RetNet/home.htm). Despite this, effective treatments for RP remain elusive due to its extreme genetic heterogeneity. Current interventions focus more on symptom management, with gene therapy being a hope for specific genotypes. There is a pressing clinical need for novel, broadly applicable therapeutic strategies that could effectively address the vast genetic diversity of RP.

The pathophysiology of RP is largely influenced by the excessive accumulation of cyclic guanosine monophosphate (cGMP) in photoreceptors (Power et al., 2020a; Zhou, Rasmussen & Ekstrom, 2021a). cGMP serves as the primary messenger molecule in the conversion of light signals into electrical signals in vertebrate photoreceptor cells. In the normal photoelectric signal transduction cycle, the degradation of cGMP by phosphodiesterase 6 (PDE6) produces GMP (Zhou, Rasmussen & Ekström, 2021). In RP, a mutation in PDE6 disrupts cGMP hydrolysis, leading to persistent cGMP accumulation in rod outer segments. This promotes unrestricted cyclic nucleotide gated channel (CNGC) access and subsequent protein kinase G (PKG) activation, resulting in an excessive calcium influx and protein phosphorylation (Paquet-Durand et al., 2009). PKG-dependent phosphorylation triggers the activation of histone deacetylases (HDACs) and poly (ADP-ribose) polymerase (PARP) (Paquet-Durand et al., 2007; Sancho-Pelluz et al., 2010b). Furthermore, calcium entry via CNGC can activate calpains, calcium-dependent cysteine proteases (Paquet-Durand et al., 2006). These processes collectively lead to photoreceptor cell death (Arango-Gonzalez et al., 2014; Sancho-Pelluz et al., 2008).

HDAC, a regulator of chromatin structure, affects the transcription of multiple genes (Yang & Seto, 2007). HDACs primarily function to remove the N-acetyl groups of acetylated lysine residues in histones and non-histones. They bind closely to negatively charged DNA, compacting chromatin and inhibiting DNA transcription, leading to gene silencing (Milazzo et al., 2020). Abnormal HDAC activity plays an important role in pathological conditions such as neurodegenerative diseases (Chuang et al., 2009), cancer (Alseksek et al., 2022; Singh, Bishayee & Pandey, 2018), and metabolic disorders (Dewanjee et al., 2021). Epigenetic regulation through HDAC inhibition can protect against inherited retinal degenerative diseases caused by diverse gene mutations, thereby preserving photoreceptor cells (Trifunović et al., 2016; Trifunović et al., 2018). Additionally, clinical trials involving HDAC modulators have demonstrated that HDAC inhibition is an effective treatment for retinal diseases, including retinal degenerative diseases (Birch et al., 2018; Chen et al., 2019).

In the cGMP-dependent cell death pathway, PARP is a significant molecule integral to photoreceptor cell death in IRDs (Paquet-Durand et al., 2007; Sahaboglu et al., 2016). The interaction between histone acetylation and PARP has been previously established (Verdone et al., 2015). Accordingly, we hypothesized that an interaction between HDAC and PARP exists in rd1 mice. Henceforth, the aim of the current research is to investigate the role of HDACs in photoreceptor cell death in a Pde6brd1 mouse model. The process of photoreceptor cell death in rd1 was effectively mitigated by the HDAC inhibitor suberoylanilide hydroxamic acid (SAHA) through the inhibition of both HDAC and PARP activities. Therefore, HDAC inhibitors may potentially serve as effective neuroprotective therapeutic agents to delay or prevent the progressive death of photoreceptor cells in RP.

Materials & methods

Animal models

The experimental subjects employed in this study consisted of C3H/HeA Pde6brd1/rd1 mice (rd1) and their congenic wild-type C3H/HeA Pde6b+/+ counterparts (wild-type, WT), with no gender preference. Initially sourced from the Cell Death Mechanism group, Institute for Ophthalmic Research, Tübingen University (Germany), these mice were housed in a specific pathogen-free environment at the Laboratory Animal Center of Yunnan University (License No. SCXK (Dian) K2021-0001). Regular genotyping was conducted on both animal colonies to confirm the presence or absence of disease-causing mutations. The mice were maintained in an environment with a 12-hour light-dark cycle and had unrestricted access to both food and water. Neonatal mice were classified as postnatal (P0) if born within 24 h, and as P1 if born after 24 h. The experimental and control groups comprised rd1 and WT mice, respectively. Throughout all procedures, strict adherence was given to the principles of animal use encompassing the 3Rs (replacement, reduction, and refinement) and the ARVO statement on the use of animals in ophthalmology and vision research. All procedures were performed in compliance with the ARVO statement for the use of animals in Ophthalmic and Visual Research. The Ethical Review Board of Yunnan University (YNU20220303, YNU20220149) has duly reviewed and granted approval for the protocols.

Single-cell RNA-seq (scRNA-seq) and bioinformatics analysis

At distinct time points of P11 (n = 3; retinas n = 6), P13 (n = 3; retinas n = 6), and P17 (n = 3; retinas n = 6), both WT and rd1 mice were euthanized. Following the approved euthanasia procedure, the mice were initially anaesthetized with avertin (1:40) delivered via intraperitoneal injection, and then decapitated to confirm death. After verifying the absence of vital signs such as heartbeat and respiration, the retinas were harvested for further examination. The eyeballs were promptly submerged in DPBS (phosphatide-buffered saline without calcium and magnesium; CAT: 21-040-CVC; Corning Inc., Corning, NY, USA), pre-chilled to 4 °C, to remove any blood or impurities. They were then incubated in 0.12% Proteinase K (539480; MilliporeSigma; Burlington, MA, USA) at 37 °C for 1 min, followed by immersion in basal medium (C11875500BT; Gibco, Thermo Fisher Scientific, Waltham, MA, USA) containing 50% fetal bovine serum (FBS; 900-108; GeminiBio, West Sacramento, CA, USA) for 2 min. The samples were finally rinsed in fresh DPBS. After removing the cornea, sclera, iris, lens, and vitreous under a microscope, the retinal tissues, consisting of retina-RPE-choroid, were fully immersed in pre-cooled MACS Tissue Storage Solution (130-100-008; Miltenyi Biotec, North Rhine-Westphalia, Germany) at 4 °C to ensure adequate activity and quantity of retinal cells for subsequent experimental investigation. Libraries were prepared using the Chromium Single-Cell 30 Reagent Kit v3 (10 × Genomics, Shanghai, China) and sequenced on an Illumina NovaSeq PE150 instrument. The Seurat package in R was employed for retinal scRNA-seq analyses. The “FilterCells” function was utilized to eliminate cells that exhibited a substantial number of outlier genes, potentially indicating the presence of polysomes, and a high proportion of mitochondrial genes, possibly suggesting cell death. Gene expression was normalized using the LogNormalization method. The dataset then underwent principal component analysis (PCA) to reduce its dimensionality, utilizing t-SNE/UMAP dimensionality reduction techniques. Cells were classified using their PCA scores with the Seurat tool. The “FindAllMarkers” function of the Seurat package was employed to identify differentially expressed genes (DEGs) for each individual cluster. The screening threshold was set at —avg_logFC—>0.58 and p < 0.05.

High-throughput CUT&Tag

The CUT&Tag assay was performed with modifications according to the previous method (Kaya-Okur et al., 2019). Frozen retinal tissues were collected from WT and rd1 mice euthanized at different time points of P11, P13, and P17. Native nuclei were isolated from the frozen samples using the previous approach (Corces et al., 2017). Once the nuclei were collected, they were gently washed twice with a wash buffer that contained 20 mM HEPES pH 7.5, 150 mM NaCl, 0.5 mM Spermidine, and 1 × Protease inhibitor cocktail. After adding 10 µL of Concanavalin A coated magnetic beads from Bangs Laboratories to each sample, they were incubated for 10 min at room temperature (RT). The unbound supernatant was discarded, and the bead-bound cells were resuspended in dig-wash buffer, which consisted of 20 mM HEPES pH 7.5, 150 mM NaCl, 0.5 mM Spermidine, 1 × Protease inhibitor cocktail, 0.05% Digitonin, and 2 mM EDTA. A primary antibody anti-HDAC (cat. no. 06-720) or normal rabbit IgG control antibody (cat. no. 12-370, Millipore, Burlington, MA, USA) was added to the resuspended cells in a 1:50 dilution and the mixture was incubated on a rotating platform overnight at 4 °C. Using a magnet stand, the primary antibody was separated and discarded. The secondary antibody, an Anti-Rabbit IgG antibody derived from Goat monoclonal (cat. no. AP132, Millipore, Burlington, MA, USA), was diluted 1:100 in dig-wash buffer and the cells were incubated at RT for 1 h. The cells were washed 2-3 times in dig-wash buffer using the magnet stand. To prepare a suitable concentration, a 1:100 dilution of pA-Tn5 adapter complex was prepared in dig-med buffer, containing 0.01% Digitonin, 20 mM HEPES pH 7.5, 300 mM NaCl, 0.5 mM Spermidine, and 1 × Protease inhibitor cocktail. The resulting solution was then incubated with cells at RT for 1 h. The cells were washed 2-3 times for 5 min each time with one mL of Dig-med buffer, and then resuspended in Tagmentation buffer, containing 10 mM MgCl2 in Dig-med buffer, and incubated at 37 °C for 1 h. DNA was purified using phenol-chloroform-isoamyl alcohol extraction and ethanol precipitation. For library amplification, 21 µL DNA was mixed with 2 µL of a universal i5 primer and a uniquely barcoded i7 primer. The addition and subsequent mixing of 25 µL NEBNext HiFi 2 × PCR Master mix took place. The sample was placed in a Thermocycler equipped with a heated lid and subjected to the following cycling conditions: gap filling at 72 °C for 5 min, denaturation at 98 °C for 30 s, followed by 14 cycles of 98 °C for 10 s and 63 °C for 30 s, then a final extension at 72 °C for 1 min, and hold at 8 °C. The libraries were cleaned using XP beads (Beckman Coulter, Brea, CA, USA). The size distribution of the libraries was determined using Agilent 4200 TapeStation analysis, and the libraries were pooled to achieve equal representation as intended, with the final concentration meeting the manufacturer’s recommendations. DNA sequencing was performed on the Illumina Novaseq 6000, using 150 bp paired-end sequencing methodology strictly according to the manufacturer’s guidelines.

Data processing and analysis of CUT&Tag

The sequencing data derived from the CUT&Tag assay was subjected to a series of bioinformatics analyses to identify the differentially accessible chromatin regions between the rd1 and WT mice. Firstly, the raw sequencing data in FASTQ format was processed using custom Perl scripts to remove low-quality reads, adapter sequences, and poly-N sequences. The resultant high-quality clean data was then used for all further analyses. The quality of the clean data was assessed by calculating the Q20, Q30, and GC content. Next, the clean reads were aligned to the mouse reference genome (GRCm38/mm10) using the Burrows-Wheeler Aligner (BWA) software. Only the uniquely mapped reads were retained and converted to BAM format for downstream analyses. Peak calling was performed using the Model-based Analysis of ChIP-Seq (MACS2) software, with a q-value cutoff of less than 0.05 to ensure the reliability of the identified peaks. The peaks were then annotated using the HOMER’s annotatePeaks.pl script, which associates the identified peaks with their corresponding genes. The differential accessibility of chromatin regions between the rd1 and WT samples was then evaluated. To do this, the peak files of all samples were first merged using the BEDTools software to create a consensus set of peaks. The number of reads within each consensus peak for each sample was then counted using BEDTools multicov. The DESeq2 R package was then used to identify the differentially accessible peak based on the read count data, with a threshold of —log2 fold change—> 1 and p-value < 0.05.

Gene ontology (GO) and KEGG enrichment analyses

The RNA-seq and CUT&Tag datasets were cross-referenced to pinpoint shared DEGs, which were subsequently subjected to enrichment analysis using GO and KEGG pathways. To elucidate the biological significance of these genes within the experiment, a comprehensive GO analysis was undertaken. This analysis facilitated the investigation of significant or representative gene profiles (Ashburner et al., 2000). GO annotations were acquired from three major databases: the NCBI (http://www.ncbi.nlm.nih.gov/), UniProt (http://www.uniprot.org/), and the Gene Ontology (http://www.geneontology.org/). The Fisher’s exact test was employed to determine statistically significant GO categories, and the resultant p-values were adjusted for multiple comparisons using the False Discovery Rate (FDR) method. Further, to identify significant biological pathways involving these genes, we employed pathway analysis based on the KEGG database. The Fisher’s exact test was used to select significant pathways, with the threshold of significance set by both the P-value and FDR.

Eyecups preparation

WT and rd1 mice were euthanized at distinct time points (P9, P11, P13, P15, and P17; six eyeballs from six different mice at each time point). After thorough washing with phosphatide-buffered saline (PBS), the eyeballs were initially fixed washing using 4% paraformaldehyde (PFA) for a duration of 10 min. Subsequently, the cornea, iris, and lens were meticulously excised under a Zeiss stereomicroscope, and the remaining eye cups were further fixed using 4% PFA for an additional 40 min. The eye cups were then washed thrice with PBS and underwent a sequential dehydration process with 10, 20, and 30% sucrose solutions. Subsequently, the eye cups were embedded in OCT Tissue-Tek agent (Sakura Finetek, Alphen aan den Rijn, Netherlands), and rapidly frozen using liquid nitrogen. These specimens were then sectioned into 10 µm slices with a CryoStar NX50 OVP cryostat (Thermo Fisher Scientific, Waltham, MA, USA). After histological processing, the sections were s air-dried at 40 °C for 45 min. Finally, they were stored at −20 °C for subsequent analysis.

Retinal explant culture procedure

At P5, six rd1 mice and six WT mice were euthanized by decapitation. Following thorough sterilization using 75% ethanol, the eyes were extracted under sterile conditions and briefly washed in basal medium for 5 min. Subsequently, the eyes were incubated for 1 min at 37 °C with 0.12% Proteinase K (539480; MilliporeSigma, Burlington, MA, USA), and then transferred to a basal medium containing 50% FBS (900-108; GeminiBio, West Sacramento, CA, USA) for 5 min to neutralize the Proteinase K. The eyes were then rinsed again in the basal medium. Careful removal of the cornea, lens, iris, vitreous body, and optic nerve from the eyeballs was performed. Following this, the retinas, along with the associated retinal pigment epithelium, were carefully detached from the sclera, divided into four wedge shapes, and relocated to a 0.4 µm Cell Culture Plate insert (3412; Transwell, 24 mm Insert, 6 Well Plate; Corning Inc., Corning, NY, USA). The retinas were then spread out like a butterfly, positioning the ganglion cell layer upwards. The membrane was then transferred to 6-well culture plates and bathed in one mL of Complete Medium (Gibco R16 medium with supplements (Belhadj et al., 2020)), and incubated at 37 °C in a humidified environment with 5% CO2. All procedures were conducted under sterile conditions.

Treatment of retinal explants with SAHA

Following extraction at P5, retinal explants were cultured in a complete medium for a period of 48 h without any intervention to acclimate to the in vitro environment. At P7, the explants were segregated into 4 groups (n = 12 each): untreated WT, WT treated with 1 µM SAHA (S1047; Vorinostat, Selleck Chemicals, Houston, TX, USA), untreated rd1, and rd1 treated with 1 µM SAHA. Throughout the cultivation phase, the nutrient solution was refreshed every 48 h, and the process was concluded on the 11th postnatal day via either fixation using 4% paraformaldehyde (n = 6 for every group) or direct freezing in liquid nitrogen without fixation (n = 6 for every group). The explants were then sectioned into 10 µm slices using a cryostat device (CryoStar NX50 OVP; Thermo Fisher Scientific, Waltham, MA, USA), after which they were dried and stored at −20 °C for future use.

Terminal deoxynucleotidyl transferase dUTP nick-end labeling (TUNEL) assay

An in-situ cell death detection kit (12156792910; In Situ Cell Death Detection Kit, TMR red; Roche, Basel, Switzerland) was employed to conduct the TUNEL assay. The fixed slides were first rehydrated in a PBS solution at RT for 15 min. Following this, they were treated with Proteinase K (0.12%) diluted in 50-mM Tris-buffered saline (TBS), which was added at a ratio of 1 µL to seven mL of TBS, and incubated for 5 min. The slides were then washed three times with Tris buffer for 5 min each and immersed in a mixture of ethanol and acetic acid (70:30) for 5 min at −20 °C. After this, the slides were washed thrice with TBS for 5 min each. They were then blocked with a solution containing 1% bovine serum albumin for 1 h at RT. Following this, the slides were incubated in the TUNEL working solution, which was composed of 10 parts blocking solution, 9 parts TUNEL labeling solution, and 1-part TUNEL enzyme, for 1 h at 37 °C. The sections were washed three times with PBS, then mounted with Vectashield containing DAPI (Vector Laboratories Inc., Burlingame, CA, USA). Images were captured using a Zeiss Imager. M2 microscopy was used for further analyses.

HDAC activity assay

The assessment of HDAC activity was conducted using 4% PFA -fixed cryosections of tissue sections, based on a modified version of the Fluor de Lys Fluorescent Assay System (Biomol GmbH, Hamburg, Germany) (Samardzija et al., 2020). Retinal sections were exposed to an assay involving 200 µM FLUOR DE LYS®-SIRT2 deacetylase substrate (BML-Kl179-0005; Enzo Life Sciences Inc., Farmingdale, NY, USA) and 2 mM NAD+ (BML-KI282-0500; Enzo Life Sciences Inc.) in assay buffer (50 mM Tris/HCl, 137 mM NaCl; 2.7 mM KCl; 1 mM MgCl2; pH 8.0) for 3 h at RT. The sections were then washed with PBS and fixed by immersion in methanol at −20 °C for 20 min. The slides were mounted using FLUOR DE LYS® developer II concentrate (1:20; BML-KI176-1250; Enzo Life Sciences Inc., Farmingdale, NY, USA) and subjected to microscopic examination.

PARP activity assay

The PARP activity assay was used to evaluate overall PARP activity within unfixed tissue sections (Belhadj et al., 2021). The retinal tissue sections were first incubated and rehydrated in a phosphate-buffered saline solution for 10 min. The segments sections were then subjected to a reaction solution composed of 10 mM MgCl2, 1 mM dithiothreitol, and 50 µM 6-Fluo-10-NAD+ (N023; Invitrogen, Thermo Fisher Scientific, Waltham, MA, USA) in 100 mM Tris buffer with 0.2% Triton X-100, pH 8.0, for a duration of 3 h at 37 °C. Following three 5-min washes in PBS, the sections were mounted in VECTASHIELD containing DAPI (Vector Laboratories Inc, Newark, CA, USA) for immediate visualization under a Zeiss Imager and an M2 microscope.

Immunofluorescence

Fixed eyecups or retinal explants slides were rehydrated in PBS for 10 min at RT. Afterward, the slides were incubated in a blocking solution composed of 10% normal goat serum and 1% BSA in 0.3% PBS-Triton X100 for 1 h at RT. The primary antibodies, anti-Cone arrestin (rabbit, 1:500, AB15282, Sigma-Aldrich, St Louis, MO, USA) and anti-Rhodopsin (mouse, 1:1,000, MAB5316, Sigma-Aldrich, St Louis, MO, USA), were diluted in the blocking solution and s incubated overnight at 4 °C. Following three washes with PBS, secondary antibodies were applied: Goat anti rabbit IgG secondary antibody, Alexa Fluor 568 (1:300, ab175471, Abcam, Cambridge, UK) and Goat anti-mouse IgG secondary antibody, Alexa Fluor 488 (1:1,000, ab150113, Abcam, Cambridge, UK), both diluted in PBS. The slides was then incubated for 1 h at RT. After subsequent washing with PBS, the slides were mounted with Vectashield containing DAPI (Vector Laboratories Inc., Burlingame, CA, USA).

Western blot

Retinas were harvested and homogenized after six days in culture. The retinal tissues underwent protein extraction using radioimmunoprecipitation assay (RIPA) lysis buffer (Beyotime, Shanghai, China). Protein concentration was subsequently measured using the BCA Protein Assay Kit (P0009, Beyotime, Shanghai, China). Ten micrograms of protein samples were equally loaded and separated on 10% polyacrylamide gels containing 0.1% sodium dodecyl sulphate (SDS). The separated samples were then transferred to polyvinylidene fluoride (PVDF) membranes. After immersion in a blocking solution of 5% BSA for 1.5 h, the membranes were incubated with primary antibodies. The primary antibodies used were Anti-acetyl-Histone H3 antibody (06-599, Sigma-Aldrich, St Louis, MO, USA), Anti-acetyl-Histone H4 antibody (06-866, Sigma-Aldrich, St Louis, MO, USA), and Anti-PARP1 Rabbit pAb (GB111501, Servicebio, Wuhan, China), all diluted at a ratio of 1:1,000 in the blocking solution. The incubation was carried out overnight at 4 °C. After three washes in TBST, the membranes were then incubated with the appropriate secondary antibodies for 1 h at RT. The bands were visualized using the ECL Plus Detection System (CAT: WBKLS0100, Immobilon Western HRP, Millipore, Germany).

Microscopy, cell counting, and statistical analysis

Fluorescence microscopy was facilitated using a Zeiss Imager M2 microscope equipped with an AxioCam 506color digital camera, and images were captured using Zeiss Zen pro 2.6 software. Images were captured from at least 6 random locations per section at 20 × magnification using the Z-stack mode of Zen 2.6. The total cell count was calculated by dividing the area of the outer nuclear layer (ONL) by the average size of individual cells. The percentage of positive cells was determined by dividing the total number of photoreceptors in the ONL by the total number of positive cells. The data presented represent the mean and standard deviation (SD) derived from a minimum of three sections for at least three distinct animals. Experimental groups were compared statistically using one-way analysis of variance (ANOVA) and Bonferroni’s correction, performed in Prism 9 for Windows (GraphPad Software, San Diego, CA, USA). The levels of statistical significance was defined as follows: not significance (n.s.); *, p < 0.05; **, p < 0.01; ***, p < 0.001; and ****, p < 0.0001.

Results

ScRNA-seq analysis of WT and rd1 mouse retinas

Approximately 18 isoforms of PARP and three classes of HDACs exist, yet their specific roles in the diverse forms of IRD remain unclear. IRDs have implicated approximately 260 genes. We conducted single-cell transcriptome analysis on retinal tissues from both wild-type (WT) and rd1 mice on P11, P13, and P17. The collected sequencing data consisted of 43,979 cells, of which 23,068 (52%) and 20,911 (48%) were from WT and rd1 retinas, respectively. We classified these cells into eight distinct categories: rod photoreceptors, cone photoreceptors, bipolar cells, amacrine cells, Müller cells, horizontal cells, microglial cells, and vascular cells (Fig. 1A). The relative abundances of WT and rd1 cells at each time point are depicted in Fig. 1B, with 5142 cells from WT P11, 12,737 from WT P13, 5189 from WT P17, 7748 from rd1 P11, 7181 from rd1 P13, and 5982 from rd1 P17. Given the strong association between HDAC and PARP activation and photoreceptor degeneration in rd1 mice, we evaluated the expression of HDAC- and PARP-related genes in rod and cone cells (Ta 1C-F). Notably, Hdac1, an important member of HDAC Class I, was highly expressed in rd1 rod cells at P17, while Parp1 was markedly downregulated, particularly in cone cells. The expression patterns of other HDAC and PARP family members varied at different time points in the rod and cone cells (Figs. S1–S2). However, immunofluorescence staining of several HDAC isoforms (HDAC1, HDAC2, HDAC9, HDAC11) revealed no significant difference between WT and rd1 retinas (Fig. S3).

Figure 1 Single-cell RNA-seq analysis of WT and rd1 mouse retinas.

(A) Distribution of eight cell clusters between WT and rd1. (B) Distribution of WT vs. rd1 cells at each time point. (C–D) Gene expression of Hdac1 in the rods (C) and cones (D) at different time points. (E–F) Gene expression of Parp1 in the rods (E) and cones (F) at different time points. WT, wild-type; HDAC, histone deacetylase; PARP, poly (ADP-ribose) polymerase. Significant differences in gene expression were determined by screening for genes with an average log fold change (—avglogFC—) > 0.58 and a p-value < 0.05.

CUT&Tag analysis for HDAC in WT and rd1 mouse retinas

The downstream mechanisms of HDAC were investigated in retinal tissues from WT (C3H) and rd1 mice at P11, P13, and P17 using the CUT&Tag assay. Peak calling statistical analyses revealed 989 upregulated and 466 downregulated HDAC peaks in WT (C3H) mice compared to rd1 mice (Table S1). Most differential peaks were located in intergenic (36.7%) and intron (31.12%) regions (Fig. 2A). We employed an integrated approach of CUT&Tag and RNA-seq analysis to identify DEGs and their respective biological functions and pathways. By using a Venn diagram, we identified 13 overlapping downregulated and 7 upregulated DEGs between RNA-seq and CUT&Tag (Fig. 2B and Table S2). GO enrichment analysis revealed that the DEGs were primarily associated with the regulation of cell death, autophosphorylation of proteins, and immune effector function (Fig. 2C). KEGG analysis presented that DEGs showed enrichment in primary immunodeficiency, adipocytokine signaling pathway, and T cell receptor signaling pathway (Fig. 2D).

Figure 2 CUT&Tag analysis for HDAC in WT (C3H) and rd1 mouse retinas.

The proportion and distribution of differential HDAC peaks in promoters, 5′-UTR, 3′-UTR, exon, intron, and intergenic regions. (B) The overlapping differentially expressed genes (DEGs) between CUT&Tag and RNA-seq analyses. (C, D) GO and KEGG enrichment analyses of DEGs.

To assess the extent of photoreceptor cell death in rd1 and WT mice over time, we performed TUNEL assays on mouse eyeballs at P9, P11, P13, P15, and P17. A significant reduction in the number of TUNEL-positive ONL cells was observed in WT retinas as compared to rd1 retinas at all time points, including P9 (0.041%), P11 (0.023%), P13 (0.025%), P15 (0.034%), and P17 (0.011%) (Figs. 3A1–3A5). In rd1 mice, the proportion of TUNEL-positive cells in the ONL of rd1 mice was low at P9 (0.046%), but this ratio surged at P11, attaining 1.175%, and reached a peak at P13 with 5.292%. Thereafter, the percentage progressively reduced at P15 (3.736%) and P17 (3.040%) (Figs. 3B1–3B5). As indicated by TUNEL staining, the onset of retinal photoreceptor cell death was evident in the early stages and exhibited a significant escalation from P9 onwards in rd1 mice. At P9, the rd1 models presented the highest number of photoreceptor rows, with 11 ONL cell rows persisting, which gradually diminished over time (Fig. 3J). Up until P17, we measured and recorded the count of photoreceptor cell rows and dying cells (Figs. 3I–3J).

Figure 3 HDAC activity paralleled to TUNEL-positive cells in ONL at different time points.

(A1-A5, B1-B5) TUNEL staining shows the dying cells (red) in the outer nuclear layer (ONL) of WT mice and rd1 mice. (C1-C5, D1-D5) HDAC activity assay shows HDAC-positive cells (white) in the ONL of WT and rd1 mice. (E1-E5, F1-F5) Immunofluorescence staining for cone arrestin (red) in WT and rd1 mice retinas. (G1-G5, H1-H5) Immunofluorescence staining for rhodopsin (green) in WT and rd1 mice retinas. DAPI (blue) was used as a nuclear counterstain. (I–M) Quantification of (I) the percentage of TUNEL-positive cells in the ONL, (J) photoreceptor row counts, (K) the percentage of HDAC-positive cells in the ONL, (L) cone cell number, and (M) rhodopsin mean fluorescence intensity at outer segment. Images shown are representative of observations for at least six different specimens of each genotype. Error bars represent SD. TUNEL, terminal deoxynucleotidyl transferase dUTP nick-end labeling; INL, inner nuclear layer; ONL, outer nuclear layer; GCL, ganglion cell layer. OS, outer segment. Scale bar = 50 µm. * p < 0.05; ** p < 0.01; *** p < 0.001; and **** p < 0.0001.

Immunofluorescence staining employing anti-cone arrestin and anti-rhodopsin antibodies was conducted to further scrutinize the temporal changes in cones and rods. At P17, an assessment of the count of surviving cone cells (Figs. 3E1–3E5, 3F1–3F5, 3L) revealed a partial loss of rd1 cones, while a reduction in the mean fluorescence intensity of rhodopsin was noted at the outer segment of rd1 rods (Figs. 3G1–3G5, 3H1–3H5, 3M). At the P17 stage in rd1, a mislocalization of rhodopsin within the ONL was discerned, attributable to aberrations in the transportation of rhodopsin to the outer segment of the rod. Such abnormalities are instigated by mutations in genes associated with retinal degeneration (Power et al., 2020a).

To ascertain the differential HDAC activity between WT and rd1 mice at various stages, we conducted an HDAC activity assay using eyecups from both mouse types at P9, P11, P13, P15, and P17. The purpose was to determine the number of HDAC-positive cells in the ONL. The ratio of HDAC-positive cells in the ONL of rd1 mice was noted to be minimal on P9 and P11, accounting for 0.03% and 0.02% respectively. However, a significant upsurge was recorded at P13, with 2.44% of HDAC-positive cells. Subsequently, a gradual decline was observed at P15 and P17, with 1.02% and 0.39% of HDAC-positive cells respectively (Figs. 3D1–3D5). In the WT group, the HDAC activity in the ONL was markedly low across all time-points (P9: 0.009%; P11: 0.006%; P13: 0.013%; P15: 0.004%; and P17: 0.008%) (Figs. 3C1–3C5), with a statistically significant variance in the rate of HDAC-positive cells between the rd1 and WT groups at P11, P13, P15, and P17 (Fig. 3K). The correlation between HDAC activity and the occurrence of TUNEL positive cells in the ONL at various time points underscores the pivotal role of HDAC in photoreceptor degeneration.

HDAC inhibitor SAHA delays photoreceptor loss in rd1 mouse retinas

To investigate the influence of SAHA, an HDAC inhibitor, on the progression of photoreceptor cell death in rd1 mice, we employed a TUNEL assay to quantify the dying cells in the ONL after administering SAHA. Additionally, we performed a quantitative analysis of ONL thickness and photoreceptor rows. SAHA exhibited a protective effect on retinal explant cultures, which was dose-dependent. At a concentration of 1 µM, the number of TUNEL positive cells was minimal and photoreceptor rows were maximally preserved (Figs. 4A6, 4B, 4C). However, toxicity was evident at higher concentrations of 10 µM and 50 µM, resulting in widespread cell death (Figs. 4A7–4A8). The proportion of TUNEL-positive cells in the ONL of rd1 retinal explants was markedly higher than that of the WT in vitro (p < 0.0001). Upon administering 1 µM SAHA, a significant decrease in TUNEL-positive cells in the ONL of rd1 mice was noted, compared to the untreated rd1 group (p < 0.0001). Conversely, no significant difference was observed between the untreated and SAHA-treated WT groups (p > 0.05). Additionally, photoreceptor rows in rd1 retinal explants were decreased compared to those in WT retinal explants (p < 0.01). SAHA treatment resulted in a significant increase in the number of photoreceptor rows in rd1 compared to the untreated rd1 group (p < 0.01). No significant difference in photoreceptor rows was observed between the untreated and SAHA-treated WT groups (p > 0.05) (Figs. 4A1–4A8, 4B, 4C). These findings suggest that HDAC inhibition delays photoreceptor degeneration in the ONL of rd1 mouse retinas.

Figure 4 SAHA treatment reduces rd1 photoreceptor cell death in vitro.

(A1-A8) TUNEL staining shows the dying cells (red) in the outer nuclear layer (ONL) of WT mice and rd1 mice after different concerntration SAHA treatment. DAPI (blue) was used as a nuclear counterstain. (A1) Untreated, control WT retinal explants. (A2) WT retinal explants after treatment with 1 µM SAHA. (A3) Untreated, control rd1 retinal explants. (A4) rd1 retinal explants after treatment with 0.01 µM SAHA. (A5) rd1 retinal explants after treatment with 0.1 µM SAHA. (A6) rd1 retinal explants after treatment with 1 µM SAHA. (A7) rd1 retinal explants after treatment with 10 µM SAHA. (A8) rd1 retinal explants after treatment with 50 µM SAHA. Quantification of (B) the percentage of TUNEL-positive cells in the ONL and (C) photoreceptor row counts. Images shown are representative of observations for at least six different specimens of each genotype. Error bars represent SD. TUNEL, terminal deoxynucleotidyl transferase dUTP nick-end labeling; INL, inner nuclear layer; ONL, outer nuclear layer; GCL, ganglion cell layer. Scale bar = 50 µm. * p < 0.05; ** p < 0.01; *** p < 0.001; and **** p < 0.0001.

Furthermore, immunofluorescence staining of cone arrestin (Figs. 5A1–5A4, 5B) and rhodopsin (Figs. 5C1–5C4, 5D) revealed a modest increase in cone cell number and mean rhodopsin fluorescence intensity at the outer segment of rd1 retinas after SAHA treatment. These results suggest the HDAC inhibitor SAHA could decelerate photoreceptor loss in rd1 mouse retina explants in vitro.

Figure 5 SAHA treatment increases the survival of rd1 photoreceptors in vitro.

(A1-A4) Immunofluorescence staining for cone arrestin (red) in WT and rd1 mice retinas after SAHA treatment. (C1-C4) Immunofluorescence staining for rhodopsin (green) in WT and rd1 mice retinas after SAHA treatment. DAPI (blue) was used as a nuclear counterstain. Quantification of (B) cone cell number and (D) rhodopsin mean fluorescence intensity at outer segment. Images shown are representative of observations for at least six different specimens of each genotype. Error bars represent SD. INL, inner nuclear layer; ONL, outer nuclear layer; GCL, ganglion cell layer; OS, outer segment. Scale bar = 50 µm. * p < 0.05; ** p < 0.01; *** p < 0.001; and **** p < 0.0001.

SAHA inhibits HDAC and PARP activity in rd1 mice

To confirm the inhibitory effect of SAHA on HDAC activity, an in situ HDAC activity assay was analysis was performed on fixed retinal tissue sections of rd1 and WT mice following SAHA treatment. HDAC activity in the ONL of the WT retina was generally found to be lower than that in the ONL of rd1 mice (p < 0.0001). Notably, post-SAHA treatment, HDAC activity was observed in the ONL of rd1 mice displayed a significant decline compared to the untreated rd1 group (p < 0.0001). On the other hand, HDAC activity in the ONL of the WT retina did not exhibit a significant change post-SAHA treatment (p > 0.05; Figs. 6A1–6A4, 6B). These observations demonstrate that SAHA administration notably curtails HDAC activity in the ONL of rd1 mice.

Figure 6 SAHA inhibits HDAC and PARP activity in rd1 mice retinas.

(A1-A4) HDAC activity assay shows the HDAC positive cells (white) in the outer nuclear layer (ONL) of WT mice and rd1 mice after SAHA treatment. (A1) Untreated, control WT retinal explants. (A2) WT retinal explants after treatment with 1 µM SAHA. (A3) Untreated, control rd1 retinal explants. (A4) rd1 retinal explants after treatment with 1 µM SAHA. (C1-C4) PARP activity assay shows the PARP positive cells (green) in the ONL of WT and rd1 mice retinas after SAHA treatment. DAPI (grey) was used as a nuclear counterstain. (B, D) Quantiûcation of (B) the percentage of HDAC-positive cells and (D) the percentage of PARP-positive cells in the ONL. (E) The expression of acetylation of Histone 3 (acH3), Histone 4 (acH4) and PARP1 after SAHA treatment by western blot. (F–H) Quantiûcation of (F) relative acH3 expression, (G) relative acH4 expression, and (H) relative PARP1 expression of WT and rd1 retinas after SAHA treatment. Images shown are representative of observations for at least six diûerent specimens of each genotype. WB images are representative for blots obtained from at least three independent experiments. Error bars represent SD. SAHA, suberoylanilide hydroxamic acid; HDAC, histone deacetylase; PARP, Poly (ADP-ribose) Polymerases; INL, inner nuclear layer; ONL, outer nuclear layer; GCL, ganglion cell layer, OS, outer segment. Scale bar = 50 µm. * p < 0.05; ** p < 0.01; *** p < 0.001; and **** p < 0.0001.

To further investigate the neuroprotective mechanism of SAHA in retinal photoreceptor cells, we evaluated alterations in PARP activity, the primary initiator of the initial response to DNA damage during photoreceptor cell death in RD. The number of photoreceptors exhibiting PARP activity in the ONL of untreated WT retinas was significantly less than that in untreated rd1 retinas (p < 0.0001). Remarkably, post-SAHA administration, a substantial decline in PARP activity was detected in the ONL of rd1 mice, in comparison to the untreated rd1 group (p < 0.0001). However, SAHA did not exhibit any significant effect on PARP activity in the retinas of WT mice (p > 0.05; Figs. 6C1–6C4, 6D). Thus, SAHA treatment also can effectively suppress the PARP activity in the ONL of rd1 mice.

Furthermore, our western blotting analysis revealed that the administration of the HDAC inhibitor SAHA led to an elevation in the acetylation of Histone 3 (H3) and Histone 4 (H4) in rd1 mice. Concurrently, the expression of PARP1 was found to be reduced in rd1 mice post-SAHA treatment (Figs. 6E–6H). These results concur with the findings from HDAC and PARP activity assays, thereby substantiating the inhibitory impact of SAHA on HDAC and PARP activity.

Discussion

In the present study, scRNA-seq revealed that different isoforms of HDACs and PARPs may have distinct roles in the process of photoreceptor cell death. As indicated by prior research, HDAC has been associated with the regulation of cell death and the immune response (Blaauboer et al., 2022; Fox & Parks, 2019). The CUT&Tag analysis consistently demonstrated that the DEGs linked to HDAC were predominantly involved in the regulation of cellular apoptosis and primary immunodeficiency. Moreover, the alterations noted in HDAC function aligned with the detection of TUNEL-positive cells in the ONL at various intervals in rd1 mice. Our results indicate that SAHA treatment enhances the survival rate of rd1 photoreceptor cells in vitro, and that it inhibits HDAC activity. We also found that SAHA inhibits PARP activity, which implies that HDAC and PARP participate in a common pathway leading to photoreceptor cell death.

Overactivation of HDACs is associated with photoreceptors degeneration in rd1 mice

Overactivation of HDACs has been closely associated with photoreceptor cell death across various models of retinal degeneration (Sancho-Pelluz et al., 2010b; Trifunović et al., 2016). The current investigation employed TUNEL staining and HDAC activity detection methods on eye tissues from rd1 and WT mice at different time points. At P13, we noted an upswing in the death of rd1 photoreceptor cells, which corresponded with an increase in the proportion of TUNEL-positive cells in the ONL. The HDAC activity was notably higher than that in the WT. Additionally, the HDAC activity in the ONL of the rd1 group was significantly higher than that of the WT in retinal explants cultured from P5 and P11. These results suggest that HDAC overactivation is implicated in photoreceptor degeneration in rd1 mice.

Histone acetylation triggers the activation of certain genes, while deacetylation leads to gene silencing (Yang & Seto, 2007). Therefore, employing HDAC inhibitors to alleviate the degree of histone acetylation can activate numerous protective genes to treat certain diseases (Milazzo et al., 2020). SAHA, also known as Vorinostat, is a member of the isohydroxamic acid HDAC inhibitors class. SAHA is a potent inhibitor of both Class I and II HDACs, acting by binding to zinc ions in the catalytic domain of the enzyme. This action leads to cellular differentiation and apoptosis, as well as cell cycle arrest (Wawruszak et al., 2021). In this study, 1 µM of SAHA was employed to treat rd1 mice retinal explants in vitro, which significantly decreased HDAC activity. Additionally, the quantity of TUNEL-positive cells in the ONL of SAHA-treated rd1 mice was significantly lower than that in the untreated group, and the thickness and rows of photoreceptors were increased, indicating that SAHA had a protective effect on photoreceptors. In WT mice, SAHA treatment did not significantly impact the ONL thickness and photoreceptor rows, nor did it substantially increase the rate of TUNEL-positive cells, indicating that SAHA presents acceptable toxic side effects on the retina. The data collected from this study indicates that excessive HDAC activation plays a crucial role in rd1 mouse photoreceptor degeneration. The inhibition of HDAC activity is a primary mechanism nderlying the protective effect of SAHA on photoreceptor survival in rd1 mice. This is attributed to the ability of SAHA to induce transcriptional regulatory changes mediated by HDAC. In addition, the survival of photoreceptor cells may be affected by various other signaling molecules, considering the potential influence of HDAC suppression on additional signaling molecules in the retina, including PARP (Tokarz, Kaarniranta & Blasiak, 2016).

Interactions between HDAC and PARP in the cGMP-dependent cell death pathway in RP

PARP, a class of enzymes with at least 17 known human variants, is responsible for the transfer of ADP-ribose to target proteins, thus forming polymeric ADP-ribose chains. This activity is critical to numerous biological processes, such as chromatin regulation, transcription, RNA biology, and DNA repair (Cohen, 2020; Morales et al., 2014; Bai, 2015) (Jubin et al., 2016; Morales et al., 2014). PARP is also capable of detecting DNA damage and promotes its repair (Ko & Ren, 2012). It has been suggested that PARP might play a key role in certain types of cell death, specifically parthanatos (Yan et al., 2021). Hyperactivation of PARP1 can lead to a build-up of PAR polymers which, in turn, can trigger the release of apoptosis-inducing factor (AIF) from the mitochondria to the nucleus, causing NAD+ andATP depletion and ultimately resulting in cell death (Fatokun, Dawson & Dawson, 2014; Wang, Dawson & Dawson, 2009). A distinctive feature of non-apoptotic cGMP-dependent cell death in IRD is an overactive PARP and the consequent accumulation of PAR (Power et al., 2020), indicating a possible interaction between cGMP signaling and parthanatos (Yan et al., 2021).

In the context of IRDs, excessive activation of HDAC and PARP has been closely linked to cGMP-induced cell death (Paquet-Durand et al., 2007; Power et al., 2020a; Sancho-Pelluz et al., 2010a). However, the exact nature of the interaction between the two within IRD models remains uncertain. PARP activity could potentially impact transcription by influencing histone H3 and H4 acetylation at specific promoter levels (Verdone et al., 2015). However, in RD mice, inhibition of HDAC activity also represses PARylation (Sancho-Pelluz et al., 2010a). Our research suggests that following HDAC inhibition by SAHA, PARP activity was significantly decreased, indicating that SAHA might mitigate the aggregation of substrate PAR by decreasing PARP activity, subsequently limiting AIF release and cell death. Additionally, the observed decrease in PARP activity upon HDAC inhibition underscores that HDAC and PARP are involved in the same cGMP-dependent photoreceptor death pathway and interact with each other.

Mechanism of SAHA-mediated inhibition of photoreceptor cell death in rd1 mice

The role of HDAC in the cGMP-dependent photoreceptor cell death pathway, and the effect of SAHA treatment on photoreceptor death in RP, are encapsulated in Fig. 7. When RD gene mutations are activated, the resulting PDE6 enzyme mutation hampers effective cGMP breakdown. This scenario leads to a consistent accumulation of cGMP in the outer segment of rods, which triggers the cyclic nucleotide-gated channel (CNG) situated in the outer segment. This results in an influx of Na+ and Ca2+ into the photoreceptors (Das et al., 2021). Moreover, cGMP-dependent activation of PKG incites HDAC overactivity; subsequent histone-mediated deacetylation leads to chromatin condensation and transcriptional inhibition, impacting the recruitment of DNA repair factors and leading to increased DNA breaks, thereby activating PARP (Power et al., 2020). PARP activation induces PAR accumulation, mitochondrial dysfunction, AIF release from the mitochondria to the nucleus, and NAD+ and ATP depletion, ultimately inducing cell death (Wang, Dawson & Dawson, 2009). The HDAC inhibitor SAHA curtails HDAC activity and promotes histone acetylation, thereby mitigating chromatin condensation, DNA damage, and PARP activation. A reduction in PAR production and NAD+ and ATP consumption contributes to maintaining mitochondrial function and intracellular ATP levels, and reduces AIF release, eventually inhibiting cell death.

Figure 7 The role of HDAC in cGMP-dependent photoreceptor cell death pathway and the neuroprotective mechanism of SAHA on photoreceptor cells in retinitis pigmentosa.

Pde6b mutation-induced cGMP accumulation activates cyclic nucleotide-gated (CNG) channels in the outer segment, leading to Na+ and Ca2+ influx and photoreceptor depolarization. Additionally, cGMP-dependent activation of protein kinase G (PKG) is associated with histone deacetylase (HDAC) activity, causing chromatin condensation and DNA breaks, which may trigger PARP activation. Excessive consumption of NAD+ by poly (ADP-ribose) polymerase (PARP) and PAR production may induce mitochondrial dysfunction, leading to ATP shortage and apoptosis-inducing factor (AIF) release, eventually resulting in cell death. Treatment with SAHA blocks HDAC activation, which decreases chromatin condensation and DNA damage, thereby reducing PARP activation, NAD+ consumption, and PAR generation. This may preserve mitochondrial function and intracellular ATP levels, thereby promoting cell survival.

Our study highlights HDACs as critical players in PDE6b mutation-induced photoreceptor degeneration and provides an effective treatment approach for RP. Although our study underscores the potential of HDAC inhibitors like SAHA in managing RP, it is paramount to consider their potential off-target effects, which stem from their broad-spectrum activity (Bolden, Peart & Johnstone, 2006). Such effects might influence numerous physiological processes and introduce unintended consequences, hence potentially impinging on the therapeutic potential of this approach (Gallinari et al., 2007). Moreover, our study shows that SAHA, a broad-spectrum HDAC inhibitor, delays photoreceptor cell death but does not completely prevent it. Therefore, the roles of other non-HDAC pathways in the cGMP-dependent cell death pathway, as well as the precise interaction mechanism between HDAC and other target molecules (such as PARP and calpain), remain to be elucidated. Moreover, it is critical to investigate the effects of employing a combination therapy approach involving various target molecules on the incidence of photoreceptor cell death.

Conclusions

To sum up, our research has demonstrated that SAHA can exert a protective effect against photoreceptor cell death in in vitro retinal explants of rd1 mice by inhibiting the activities of HDAC and PARP. This research provides evidence supporting the role of HDAC in hereditary retinal degeneration. Additionally, it suggests that the HDAC inhibitor SAHA may have potential therapeutic applications in the treatment of RP. However, challenges like efficient drug delivery and defining optimal combinations of HDAC and PARP inhibitors warrant further research to ensure optimal retinal protection and therapeutic efficacy.

Supplemental Information

Supplemental Information 1 Raw data

Click here for additional data file.

Supplemental Information 2 Gene expression of HDAC family at different time points in rods and cones by single-cell RNA-seq analysis

Hdac2 (A1 in rods A2 in cones), Hdac3 (B1 in rods, B2 in cones), Hdac4 (C1 in rods, C2 in cones), Hdac5 (D1 in rods, D2 in cones), Hdac6 (E1 in rods, E2 in cones), Hdac7 (F1 in rods, F2 in cones), Hdac8 (G1 in rods, G2 in cones), Hdac9 (H1 in rods, H2 in cones), Hdac10 (I1 in rods, I2 in cones), and Hdac11 (J1 in rods, J2 in cones) of different time points (P11, P13 and P17) of WT and rd1 are indicated in the violin plot.

Click here for additional data file.

Supplemental Information 3 Gene expression of PARP family at different time points in rods and cones by single-cell RNA-seq analysis

Parp2 (A1 in rods A2 in cones), Parp3 (B1 in rods, B2 in cones), Parp4 (C1 in rods, C2 in cones), Parp6 (D1 in rods, D2 in cones), Parp8 (E1 in rods, E2 in cones), Parp9 (F1 in rods, F2 in cones), Parp10 (G1 in rods, G2 in cones), Parp11 (H1 in rods, H2 in cones), Parp12 (I1 in rods, I2 in cones), Parp14 (J1 in rods, J2 in cones), and Parp16 (K1 in rods, K2 in cones) of different time points (P11, P13 and P17) of WT and rd1 are indicated in the violin plot.

Click here for additional data file.

Supplemental Information 4 Immunofluorescence staining of several HDACs isoform (HDAC1, HDAC2, HDAC9, HDAC11)

(A1-A5, B1-B5) HDAC1 positive cell (red) in WT and rd1 mice at different time points. (C1-C5, D1-D5) HDAC2 positive cell (red) in WT and rd1 mice at different time points. (E1-E5, F1-F5) HDAC9 positive cell (red) in WT and rd1 mice at different time points. (G1-G5, H1-H5) HDAC11 positive cell (red) in WT and rd1 mice at different time points.DAPI (blue) was used as a nuclear counterstain. Images shown are representative of observations for at least three different specimens of each genotype. INL, inner nuclear layer; ONL, outer nuclear layer; GCL, ganglion cell layer. Scale bar = 50 µm.

Click here for additional data file.

Supplemental Information 5 Differentially Expressed HDAC Peaks in WT (C3H) and rd1 Mice at P11, P13, and P17

Click here for additional data file.

Supplemental Information 6 The list of overlapping 13 downregulated and 7 upregulated DEGs between RNA-seq and CUT&Tag analyses

Click here for additional data file.

Supplemental Information 7 Western blot - B-actin

Click here for additional data file.

Supplemental Information 8 Western blot - PARP1

Click here for additional data file.

Supplemental Information 9 Western blot - acH3

Click here for additional data file.

Supplemental Information 10 Western blot - acH4

Click here for additional data file.

Supplemental Information 11 Inhibition of the MAPK/c-Jun-EGR1 Pathway Decreases Photoreceptor Cell Death in the rd1 Mouse Model for Inherited Retinal Degeneration

Click here for additional data file.

Supplemental Information 12 Single-Cell Transcriptomic Profiling in Inherited Retinal Degeneration Reveals Distinct Metabolic Pathways in Rod and Cone Photoreceptors

Click here for additional data file.

Supplemental Information 13 ARRIVE LIST

Click here for additional data file.

We thank Jiayin Biotechnology Ltd. (Shanghai, China), for the assistance and suggestion with scRNA-seq assay and high-throughput CUT&Tag experiments.

Additional Information and Declarations

Competing Interests

Author Contributions

Animal Ethics

DNA Deposition

Data Availability

The authors declare there are no competing interests.

Yujie Dong performed the experiments, prepared figures and/or tables, and approved the final draft.

Jie Yan performed the experiments, analyzed the data, prepared figures and/or tables, and approved the final draft.

Wenrong Xu performed the experiments, analyzed the data, prepared figures and/or tables, and approved the final draft.

François Paquet-Durand conceived and designed the experiments, authored or reviewed drafts of the article, and approved the final draft.

Zhulin Hu conceived and designed the experiments, authored or reviewed drafts of the article, and approved the final draft.

Kangwei Jiao conceived and designed the experiments, authored or reviewed drafts of the article, and approved the final draft.

The following information was supplied relating to ethical approvals (i.e., approving body and any reference numbers):

The protocols were reviewed and approved by the Ethical Review Board of Yunnan University (YNU20220303,YNU20220149).

The following information was supplied regarding the deposition of DNA sequences:

The Single-Cell RNA-seq (scRNA-seq) is available at GEO: GSE212183.

https://www.ncbi.nlm.nih.gov/geo/query/acc.cgi?acc=GSE212183

The following information was supplied regarding data availability:

The raw data is available in the Supplemental File.

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
