# Peer review of "HDAC inhibition delays photoreceptor loss in Pde6b mutant mice of retinitis pigmentosa: insights from scRNA-seq and CUT&Tag"

_PeerJ, doi:10.7717/peerj.15659_

## Round 0.1 · original submission · Minor Revisions

Please revise your manuscript according to the comments of our Reviewers.

Reviewer 1 ·

Basic reporting

The manuscript is generally well-written and clear. The background and purpose of the study are adequately introduced. However, I recommend the following improvements to enhance clarity and readability:

Please consider revising the abstract to improve its clarity and readability. Specifically, streamline the methods and results sections in the abstract to focus on key information, and make sure the abstract adheres to the PeerJ journal format.

In the introduction, provide a brief overview of the current treatment options for RP, and highlight the unmet need for novel therapeutic strategies.

Add more recent relevant literature to the introduction and discussion sections to provide a comprehensive context for your study.

The manuscript would benefit from a thorough language and grammar check to improve readability and clarity.

Experimental design

The experimental design is appropriate and well-described.

Validity of the findings

The findings appear to be valid and well-supported by the data. However, the following points should be addressed:

Discuss the potential off-target effects of SAHA or other HDAC inhibitors and the implications of these effects on the study's conclusions.

Additional comments

The authors should provide a more in-depth discussion of the limitations of the study, including potential off-target effects of SAHA and the use of a single concentration of SAHA.

The authors should discuss the potential therapeutic implications of their findings for the treatment of retinitis pigmentosa, including the potential use of HDAC inhibitors as a therapeutic strategy, challenges associated with drug delivery, and the need for further research to determine the optimal combination of HDAC and PARP inhibitors for retinal protection.

Reviewer 2 ·

Basic reporting

no comment

Experimental design

no comment

Validity of the findings

no comment

Additional comments

In the current study, the authors investigate the roles of different isoforms of HDACs and PARPs in the process of the photoreceptor cell death. In addition, the authors found that SAHA, one HDAC inhibitor, can provide protective effects against the photoreceptor cell death. Overall, these are the novel observations, and the data support the main conclusion. However, there are minor concerns that need to be addressed, which are listed as follows:

1. Please keep the consistent name of proteins or genes, such as “CGMP” or “cGMP”.

2. Please double check the information and citation of “Verdone et al. 2015” is correct. The reviewer cannot see the same author from this reference with the current manuscript.

3. Is there any significant difference in Fig 1C-1F? If yes, please indicate the significance in the Figures and describe the statistical methods used.

4. It is better that the authors can show the list of 13 downregulated and 7 upregulated DEGs in supplemental Table or list.

5. In line 472, please delete the sentence “Add your discussion here”.

·

Basic reporting

This artical "HDAC inhibition delays photoreceptor loss in Pde6b mutant mice of retinitis pigmentosa: insights from scRNA-seq and CUT&Tag" presents a unique opportunity for a comprehensive analysis of the development of single cell gene expression and CUT&TAG in Pde6b mutant mice. The authors hoped to identify candidates such as HDACs and PARPs that are useful biomakes, and indeed this analysis can provide some new insights into the pathogeneis of RD.

Experimental design

no comment.

Validity of the findings

no comment.

Additional comments

no comment.

---

## Round 0.2 · accepted · Accept

After reviewing the authors' revised version, I believe this revised paper was significantly enhanced and improved and the concerns of the 3 Reviewers were adequately addressed. This article is ready to be considered for publication in this Journal.